# Measurement of Oxygen Transfer Rate and Specific Oxygen Uptake Rate of h-iPSC Aggregates in Vertical Wheel Bioreactors to Predict Maximum Cell Density Before Oxygen Limitation

**DOI:** 10.3390/bioengineering12040332

**Published:** 2025-03-22

**Authors:** James Kim, Omokhowa Agbojo, Sunghoon Jung, Matt Croughan

**Affiliations:** Bioprocess R&D, PBS Biotech Inc., 4721 Calle Carga, Camarillo, CA 93012, USA; oagbojo@pbsbiotech.com (O.A.); sjung@pbsbiotech.com (S.J.); mcroughan@pbsbiotech.com (M.C.)

**Keywords:** oxygen transfer rate, oxygen uptake rate, stem cells, vertical wheel bioreactors, iPSCs, maximum cell density

## Abstract

The prediction of the cell yield in large-scale bioreactor culture is an important factor for various cell therapy bioprocess operations to ensure consistency in cell quality and efficient use of resources. However, the shear sensitivity of cells used in cell therapy manufacturing can make such predictions difficult, particularly in large-scale suspension cultures that have significant stresses without representative scale down models. The PBS Vertical-Wheel (VW) bioreactors have been demonstrated to provide a homogeneous hydrodynamic environment with low shear for cell culture at various scales (0.1–80 L) and is thereby employed for various shear-sensitive cells. In this study, the oxygen transfer rate for surface aeration for three large-scale VW bioreactors was measured along with the specific oxygen uptake rate (sOUR) of iPSCs cultured in the bioreactors. The oxygen mass transfer coefficient was measured in PBS-3/15/80 L bioreactors at different agitation rates, headspace gas flowrates, and working volumes using the static gassing-out method. The sOUR of iPSCs was measured using the dynamic method in the PBS-0.1 L Mini with a custom DO probe configuration. The results from both experiments were combined to calculate the theoretical maximum cell density before oxygen limitation across VW bioreactors at 2 L/3 L/10 L/15 L/50 L/80 L working volumes at a different agitation speed and aeration rate.

## 1. Introduction

Over the years, human pluripotent stem cells have gained a tremendous amount of attention for their ability to differentiate into a variety of cell types—an ability that could be applied to a range of therapies such as pancreatic beta islet cells for Type 1 diabetes [1], thymus cells for immune disorder [2], NK cells for cancer immunotherapy [3], and many more [4,5,6]. However, to fully realize the potential of these stem cell therapeutic products, large numbers of cells are needed to supply many of the clinical trials required by international regulatory agencies as well as the subsequent commercial markets. As a result, many stem cell therapy manufacturers want to switch from static to suspension culture, wherein cells can be grown in larger volumes with continuous monitoring and feedback control of the cultural environment [7]. The Vertical-Wheel^®^ bioreactor (PBS Biotech Inc., Camarillo, CA, USA) has been proven to provide a uniform and scalable hydrodynamic environment across multiple scales [8] and hence, supports linearly scalable growth for shear sensitive cells like iPSCs, MSCs, and others [9,10].

In bioreactor cultures, gas transfer efficiency is often manipulated by either changing the agitation speed of the impeller or changing the aeration rate either through headspace gassing or sparging [11]. However, certain suspended cells—especially those grown in aggregates or on the surface of microcarriers—are sensitive to shear stress. For such cells, the agitation rate often has an impact on the quality and process metrics of the cells (e.g., size of intermediate cell aggregates and/or ultimate differentiation state of the cell product). Therefore, the range of agitation speeds or sparging rates that can be used are limited and often based on the cell type [12]. As a result, there have been several publications focused on determining suspension cell culture parameters to maintain optimal ranges of hydrodynamic forces and gas transfer using various modeling and experimental methods [8].

Nevertheless, the provision of sufficient gases—like oxygen—to cells in culture remains an important topic for large scale bioreactors. The oxygen mass transfer coefficient (k_L_a) is the parameter that describes the rate at which oxygen transitions from the gas phase into the liquid phase. This parameter directly correlates to the efficiency of oxygen delivery to cells within a bioreactor. The measurement of k_L_a in bioreactors and fermenters has been performed using a range of methods and can be found in many articles. Most of these methods involve measuring the change in dissolved oxygen (DO) percentage within the culture medium while changing the supplied gas composition [13,14]. Proactively analyzing these empirical k_L_a data and incorporating these results in the design of experiments for large scale bioreactors can avoid hindered cell growth due to oxygen limitation.

Another important factor for large-scale bioreactor cell culture is the specific oxygen uptake rate (sOUR) of the cells. Different cells have different sOUR; therefore different optimal cell culture conditions are required, including availability of oxygen in the medium [15]. Several methods for sOUR measurement have also been explored extensively in the literature, ranging from oxygen-binding fluorescent reagents to the Lab-on-a-chip approach [15,16]. However, it is also important to consider the cell culture methods as it is suggested by various others in the field that different suspension cell culture modes such as aggregate or microcarrier cultures can also have an impact on sOUR as the cell morphology will be different [17,18,19].

As an initial study regarding aeration in vertical-wheel bioreactors, this investigation is limited to surface aeration. Surface aeration is frequently used for cell cultures up to 1–3 L in volume with cell densities up to 1–3 million cells per ml. These scales and cell densities are commonly seen in early seed train cultures for monoclonal antibody production and more recently in many production cultures for cell therapy. Surface aeration has been studied in the past [11] but not for vertical wheel bioreactors. Therefore, we measured the k_L_a of surface aeration for vertical wheel bioreactors with culture volumes of 2 L, 3 L, 10 L, 15 L, 50 L and 80 L by varying both agitation and headspace gas flowrate, which are the two most common methods used in the field to manipulate the k_L_a. The sOUR of iPSCs in aggregate culture were measured as well in PBS Mini bioreactors (0.1 L) with a custom DO probe configuration. Using both the k_L_a and sOUR data, the theoretical maximum cell density before oxygen limitation was calculated for a variety of culture volumes and operating conditions. This overall approach of using sOUR and k_L_a data to calculate the maximum cell density can be applied to wide range of suspension cell culture operations.

## 2. Materials and Methods

### 2.1. Bioreactor Set-Up

Vertical-Wheel^®^ bioreactors (PBS Biotech Inc. Camarillo, CA, USA) were used for this study and filled with RO water to different working volumes. Volumes of 2 L and 3 L were tested in a PBS-3 bioreactor. Volumes of 10 L and 15 L were tested in a PBS-15 bioreactor. Volumes of 50 L and 80 L were tested in a PBS-80 bioreactor. The bioreactor and vessel set-up followed the manufacturer’s protocol. The DO probes (Broadley-James. Irvine, Irvine, CA, USA) were installed in the bioreactors and calibrated according to the manufacturer’s protocol. During calibration, DO measurement of the ambient air was set as 100% DO and the reading with the DO probe unplugged was set as 0% DO.

### 2.2. k_L_a Measurement

The general principle of k_L_a measurement can be found in many articles and book chapters [20,21] and therefore will be described here in only limited detail. For this study, the static gassing out method was used to measure k_L_a. This approach relies on Equation (1), where C* represents the saturation DO percentage of liquid in equilibrium with air, C is the variable DO% of the liquid in the bioreactor and C_0_ is the saturation DO percentage of liquid in equilibrium with nitrogen.(1)$$-k_{L}a*time=ln\left(\frac{C^{*}-C}{C^{*}-C_{0}}\right)$$

The following series of steps were taken to manipulate the DO % within the bioreactor as shown in Figure 1a. First, nitrogen was introduced into the vessel to strip oxygen from the liquid in the bioreactor. (Step 1) Once the DO level stabilized at approximately 0% DO, compressed air was introduced into the vessel (Step 2) through the headspace, and the rate of change in DO was measured until the DO level stabilized at approximately 100% DO (step 3). To determine the empirical k_L_a values using Equation (1), the measured C*, C and C_0_ values were used to calculate the right-hand side of Equation (1), then plotted against time (Figure 1b) which will yield a straight line of slope (−k_L_a) between 20% and 80% DO (black dotted box in Figure 1b). A wide range of operating conditions such as different headspace gas flow rates and agitation rates were tested to fully characterize their impact on the bioreactor’s k_L_a.

### 2.3. Static and Suspension iPSC Culture for sOUR Measurement

The h-iPSC line TC1133—derived from umbilical cord blood cells (Lonza, Walkersville, MD, USA)—was used for this study. Static cell expansion prior to bioreactor culture was performed as described in detail by Dang et al. [8]. Briefly, h-iPSCs were inoculated into Nunc T-175 flasks (Thermo Scientific, Waltham, MA, USA) coated with Matrigel (Corning, Corning, NY USA) at a density of 15,000 cells/cm^2^. The cells were inoculated using mTeSR1 medium (Stemcell Technologies, VC, Canada) supplemented with 10 μM of Y-27632 (Stemcell Technologies, VC, Canada) on Day 0 and placed in an incubator maintained at 37 °C and 5% CO_2_. Daily medium exchanges with Y-27632-absent mTeSR1 medium were performed every 24 h for a culture period of 3 days (0.32 mL/cm^2^ medium working volume). To pass cells from static culture, Accutase (Stemcell Technologies, VC, 07922) supplemented with 10 µM Y-27623 was used to dissociate the cells for 7 min. The cells were then diluted in a 2:1 ratio of mTeSR1 medium supplemented with 10 μM of Y-27632 to Accutase. Single cells were collected in conical tubes, centrifuged at 500× *g* for 5 min, and resuspended in fresh mTeSR1 medium supplemented with 10 μM Y-27632. Viable cell density was measured using NucleoCounter NC-200 (ChemoMetec, Cambridege, MA, USA) and cells were inoculated into a CellSTACK-5 at 5000 cells/cm^2^ using the same protocol as described.

After a second static passage in the CellSTACK-5, the cells were harvested as previously described [8] and inoculated into a PBS-3 bioreactor batched with 95% working volume (2850 mL) of E8 based medium supplemented with 10 uM Y-27632. They were maintained at 24 RPM, 50% DO, 5% CO_2_ for a culture period of 7 days. Approximately 50% medium exchanges were performed every 24 h from day three onward with Y-27632-absent E8 medium. To perform the medium exchange, fresh medium was slowly introduced to the bioreactor at 1.04 mL/min and the spent medium was slowly removed from the bioreactor at the same rate to achieve 50% medium exchange. On day 7, aggregates were removed from the bioreactor for sOUR measurements in the PBS 0.1 mini.

### 2.4. sOUR Measurement

Various methods exist in the literature to measure the sOUR of cells [15,16]. For this experiment, the dynamic method, where the respiratory activity of the cells is measured without any gas supply into the reactor, was used to measure the sOUR of iPSC aggregates. The D145 (12 mm) DO probe (Broadley James Inc. Irvine, CA, USA) connected to an external PBS-3 bioreactor was sterilely inserted into a PBS 0.5 mini containing DMEM/F-12 (Corning, Corning, NY, USA) and placed in an incubator (37 °C, 5% CO_2_) two days prior to the sOUR measurement to ensure sterility and sensor functionality and stability at 95% DO. An amount of 140 mL of culture medium containing 280 million cells in aggregates was transferred from the PBS-3 bioreactor culture to a PBS-0.1 to completely fill the bioreactor. Once the aggregates were transferred to the PBS-0.1 mini, the calibrated sterile DO probe was moved from the PBS-0.5 into PBS-0.1 containing aggregates. Then, the change in DO was measured as shown in Figure 2a. Appropriate adapters (McMaster Carr, Santa Fe Springs, CA, USA) were used to ensure that there was no gas exchange between the inside of the PBS-0.1 Mini and the incubator environment as shown in Figure 2b. The rate of change in DO was converted into sOUR using Equation (2) (Calculation shown in Appendix A). Of the aggregate samples, amounts of 1 mL were taken before and after sOUR measurement and dissociated into single cells using Accutase to measure the cell density and viability using the NC-200 (ChemoMetec, Cambridge, MA, USA).(2)$$-sOUR*Cell\;density=\left(\frac{dC}{dt}\right)$$

## 3. Results

### 3.1. kLa of PBS 3, 15 and 80 Bioreactor

Using the static gassing-out method described above, k_L_a values for the PBS-3, -15 and -80 bioreactors at different agitation rates, headspace gas flow rates, and working volumes were measured and are shown in Figure 3. (N = 1 and the individual measured k_L_a values can be found in Appendix A). From the data provided, it appears that the changes in the agitation rate and working volume have a greater impact on the k_L_a values than changing the bioreactor scale and headspace gas flowrate.

Similar studies have been conducted by others in the field using surface aeration to measure oxygen transfer rate, such as plotting the Sherwood number against impeller Reynolds number [20], k_L_a against power factor [21], OTR against aeration rate [22] and many more. To provide similar perspective, the individual k_L_a values obtained from this study were plotted against agitation rate converted into EDR [m^2^/s^3^], main gas flow rate converted into aeration rate [vvm] and the volume ratio against 100% working volume [%] in Figure 4 The independent variables have been normalized against standard score for comparison against each other. Energy dissipation rates are given in power/mass and were determined via published power number correlations for the vertical-wheel bioreactors [23]. ANCOVA test has been performed against the measured k_L_a values with commercial statistical analysis software (DATAtab (2025)) and the test provided *p*-value of <0.001 for change in agitation rate and working volume and 0.143 for change in aeration rate. The measured slope in Figure 4 shows 0.83, −0.74 for the change in agitation rate and working volume and 0.42 for aeration. (Note: R^2^ values were 0.16, 0.27 and 0.09.)

### 3.2. sOUR Measurement

The sOUR of iPSCs in aggregate forms were measured using the dynamic method. Based on the cell numbers, the measured rate of change in DO and assuming the 0.21 mmol/L at 100% air saturation at 37 °C [24], sOUR calculated for TC1133 h-iPSC aggregates on day 7 of their culture was 17.7 amol cells^−1^ s^−1^. (Calculation shown in Supplemental Appendix A). The change in cell concentration before and after the sOUR measurement based on NC-200 counts was 3%, so no adjustment was made to the number of cells for sOUR calculation. Comparable values found in the literature for hESCs were 1–112 amol cells^−1^ s^−1^ [15].

### 3.3. Maximum Number of Cells Before Oxygen Limitation

By equating the oxygen uptake rate (OUR) and the oxygen transfer rate (OTR) in a bioreactor system (Equation (3)), a prediction for the maximum cell density can be calculated before oxygen limitation is reached (X) [25]. Several assumptions were made for this calculation. It was assumed that pure oxygen flowed into the headspace. It was also assumed the flow rate was sufficient to adequately strip out carbon dioxide made by the cells, avoid excessive accumulation of carbon dioxide in the culture fluid, and avoid significant buildup of carbon dioxide in the head space gas (i.e., a few percent by volume at the most). The DO levels across the bioreactor volume were assumed to be homogeneous (well-mixed bioreactors). All O_2_ introduced into the culture medium was assumed to be immediately consumed by the cells. The culture condition within the medium (glucose/lactose levels, pH, etc.) has been optimized to achieve optimum cell growth. The measured sOUR values of the iPSC aggregates were assumed to be constant throughout the culture. Once the cell density reaches the calculated value X, cells will stop growing due to absence of enough oxygen for all cells to grow. With these assumptions, the cell density (X) from Equation (4) will be the maximum cell density before oxygen limitation, where C* is the saturated concentration of oxygen within the medium and C is the actual concentration of O_2_ within the medium.(3)OUR=OTR(4)$$\;\;\;\;\;\;\;\;\;\;sOUR*X=k_{L}a*\;(C^{*}-C)$$

For example, when a PBS 3 bioreactor with 3 L WV, 30 rpm and oxygen is supplied at 0.3 L with 50% DO level maintained, the empirical k_L_a value is 0.438 h^−1^. Using C* of 1 mmol/L (476.19% DO within the medium), C of 0.11 mmol/L (50% DO maintained within the culture medium) as the concentration gradient, and 17.7 amol cells^−1^ s^−1^ obtained from the sOUR measurement, X from Equation (4) will equate to 4.65 × 10^6^ cells/mL. It was assumed that the solubility of oxygen in culture medium is 0.22 mmol/L at 100% air saturation [24].

In our previous publications, we investigated the impact of the agitation rate on the growth rate of iPSC aggregates in suspension culture based on energy dissipation rate (EDR) and Power/Volume (P/V) calculated empirically and through CFD simulations [8,23,25]. The determined optimum range of agitation rate for iPSCs aggregate suspension culture was 20–30 rpm for the PBS-3, 10–20 rpm for the PBS-15, and 6–12 rpm for the PBS-80. Using Equation (3), values for X with these optimum agitation rates (favorable hydrodynamic culture conditions) for iPSCs cultured in VW bioreactor are summarized in Table 1. The corresponding total number of cells (X × WV) are presented in Figure 5. The dotted black box denotes the region that provides best hydrodynamic environment for iPSC aggregate culture within VW bioreactors. The calculated X values and the total number of cells for all operating conditions from this study are provided in Appendix A

## 4. Discussion

### 4.1. Impact of Agitation Rate, Main Gals Flow Rate and Working Volume on k_L_a

As mentioned in Section 1, the k_L_a is directly related to the efficiency with which oxygen is transferred from the gaseous phase into the liquid phase. The most effective method of increasing k_L_a is sparging O_2_ bubbles directly into the medium. However, due to the hydrodynamic sensitivity, as well as the possible impacts of the protective agent Pluronic F-68 on the resulting cell properties, such options are not available for certain cell types [12]. Therefore, manipulation of variables such as agitation rate, headspace gas supply, or bioreactor working volume are often the only methods available to increase bioreactor k_L_a in the cell therapy industry unless an external oxygenator is used.

Based on the results from Figure 3, decreasing the working volume or increasing the agitation rate, rather than increasing headspace oxygen supply, are the most effective methods to increase k_L_a. However, as previously mentioned, different cell types, especially cell aggregates, have different optimal hydrodynamic conditions for optimal growth and quality. Moreover, agitation rate is often served as a controllable process input variable to manipulate the size of aggregates. These aspects necessitate careful choice for k_L_a strategy for successful cell culture and bioprocess development. In 2011, Chalmers et al. provided the summary of reported cell damage at different energy dissipation rate for various cells such as CHO, hybridoma, HeLa, mouse myeloma, etc., [26]. One can track similar data for other cell types (iPSC aggregates, MSCs on microcarrier, etc.) and establish an ideal range of agitation rates and other process attributes to maximize oxygenation and k_L_a [27,28,29].

### 4.2. Benefit of Combining the OTR and OUR to Predict Maximum Cell Density Before Oxygen Limitation

Using Equation (3), the VW bioreactor users can calculate the theoretical maximum cell density before oxygen limitation using the either empirical or literature sOUR values of their specific cell-lines. These calculations can help in the design of cell culture experiments to align with the target goal of each laboratories. For example, if 300 million iPSCs are needed for a single dosage with 100 patients in a clinical trial, based on Figure 5 and Appendix A, the operational conditions that corresponds to 3 × 10^10^ cells are PBS-3 at 2 L working volume with 50 rpm with 0.1 LPM oxygen supply, 3 L working volume at 50 rpm with 0.4 LPM oxygen supply, PBS 15 at 10 L working volume at 10 rpm with 0.2 LPM oxygen supply, PBS 15 at 15 L working volume at 20 rpm with 0.1 LPM oxygen supply. Given the media cost, the immediate choice might be PBS 3 with 2 L WV at 50 rpm agitation rate with 0.1 LPM oxygen supply. However, our recent publication indicates that 20–30 rpm agitation rate provides optimal growth environment for iPSC aggregates in PBS 3 bioreactor and 10–20 rpm for PBS 15 bioreactors [8]. Therefore, PBS 15 at 10 L WV at 10 rpm with 0.2 LPM air supply will be the best option to obtain 30 billion iPSCs.

## 5. Conclusions

The results in this paper show that the combination of k_L_a values and the sOUR of iPSCs measured in vertical wheel bioreactor can be translated to theoretical maximum cell density before oxygen limitation. This approach could be applied to a wide range of cell therapy manufacturing scenarios where growing number of cells at a target density within single cell culture run is very important to ensure consistency in cell quality. The work provided in this paper can serve as a guide to those working in the cell therapy manufacturing field to determine the necessary bioreactor and operating conditions to achieve their goals.

For future works, the sOUR of iPSCs and other cell types will be measured in the vertical wheel bioreactor in various forms, such as in different aggregate sizes, on microcarriers, on different culture days, etc., to fully capture the varying degrees of sOUR of cells as mentioned in the literature. The theoretical maximum cell density before oxygen limitation can be expanded to other cell types and will be compared with experiment results.

## Figures and Tables

**Figure 1 bioengineering-12-00332-f001:**
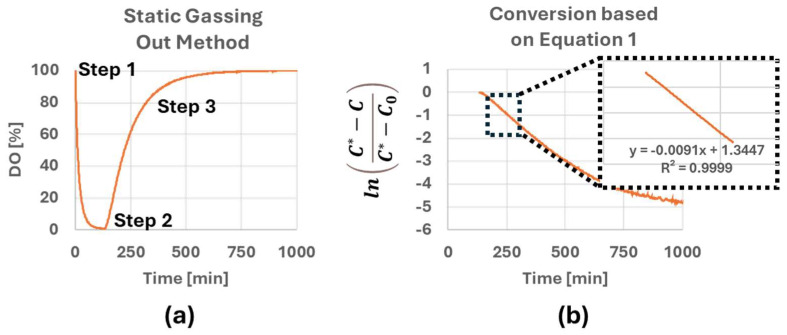
(**a**) Change in DO level inside the PBS 3 bioreactor during k_L_a measurement studies. (**b**) k_L_a calculated based on Equation (1).

**Figure 2 bioengineering-12-00332-f002:**
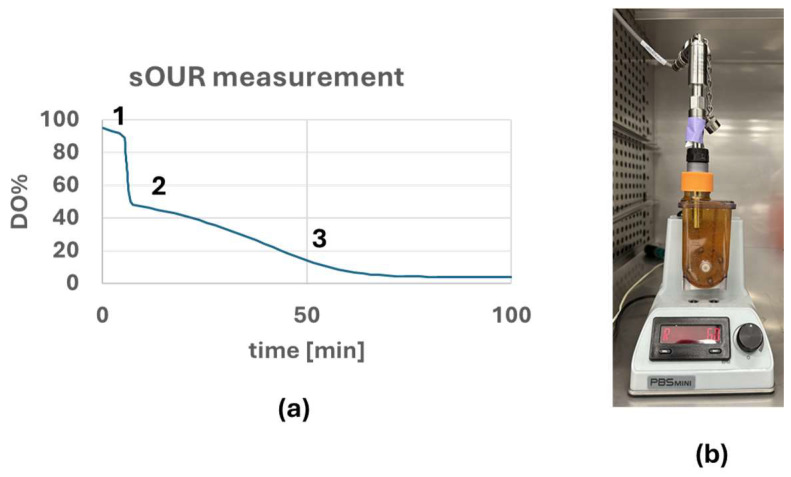
(**a**) Cells were inoculated at step 1, DO level quickly dropped to 50% (cells were cultured in 50% DO medium in PBS 3 bioreactor prior to inoculation as mentioned in Section 2.3) at step 2, and O_2_ was consumed by the cells at step 3. (**b**) Configuration of the gas-tight PBS 0.1 mini for the measurement of change in DO with iPSC aggregates inside.

**Figure 3 bioengineering-12-00332-f003:**
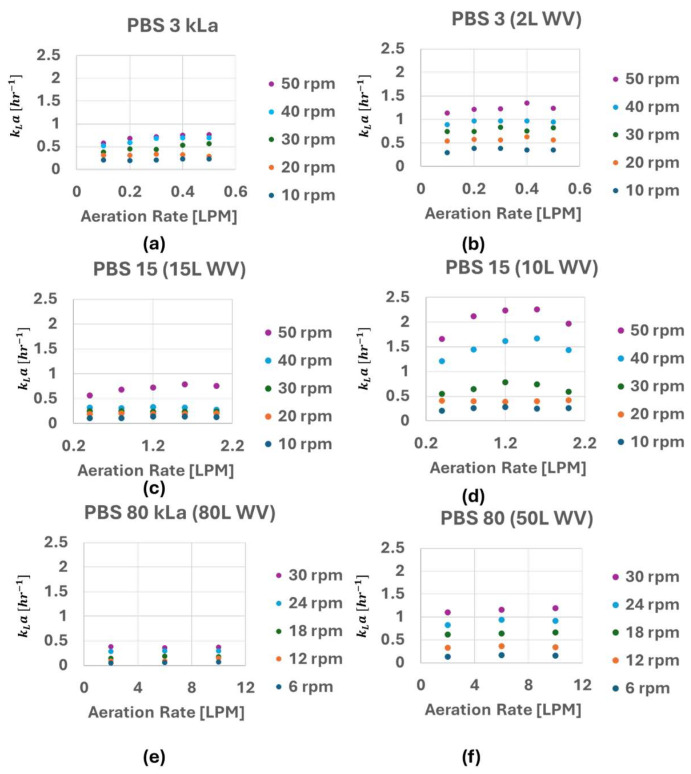
Summary of the k_L_a values measured with the PBS-3 bioreactor at 3 L and 2 L WVs, the PBS-15 at 15 L and 10 L WVs, and the PBS-80 at 50 L and 80 L WV (**a**–**f**).

**Figure 4 bioengineering-12-00332-f004:**
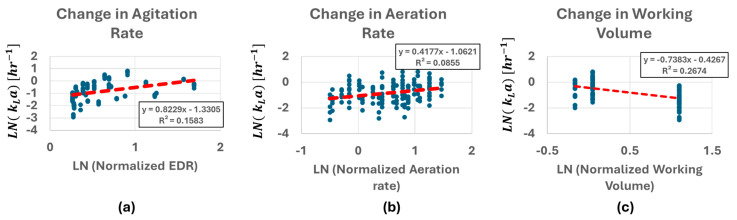
The effect of energy dissipation rate, aeration rate, and the working volume (**a**–**c**) on the measured k_L_a values.

**Figure 5 bioengineering-12-00332-f005:**
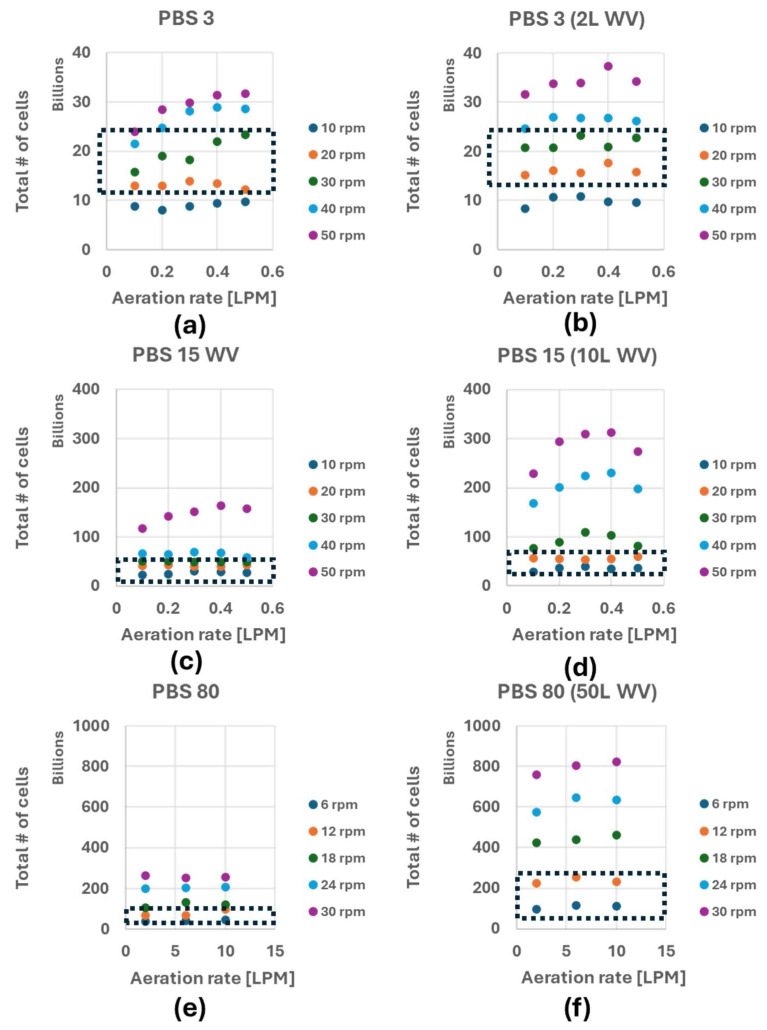
The total number of cells obtainable based on the theoretical maximum cell density before oxygen limitation (X) in PBS-3 bioreactor at 3 L and 2 L WVs, the PBS-15 at 15 L and 10 L WVs, and the PBS-80 at 80 L and 50 L WV (**a**–**f**). The black dotted box denotes region of optimum hydrodynamic condition for the cells determined empirically.

**Table 1 bioengineering-12-00332-t001:** Maximum cell density before oxygen limitation (X) summarized for favorable iPSCs hydrodynamic culture conditions within VW bioreactor.

	PBS 3 2 L WV	PBS 3 3 L WV	PBS 15 10 L WV	PBS 15 15 L WV	PBS 80 50 L WV	PBS 80 80 L WV
X at lower rpm * [cells/mL]	6.64 × 10^−6^	4.32 × 10^−6^	2.91 × 10^−6^	1.49 × 10^−6^	1.91 × 10^−6^	7.47 × 10^−5^
X at higher rpm * [cells/mL]	1.14 × 10^−7^	7.81 × 10^−6^	5.90 × 10^−6^	2.82 × 10^−6^	5.07 × 10^−6^	1.99 × 10^−6^

* The lowest and highest rpm within the optimal agitation range for each bioreactor.

## Data Availability

All of the experimental data are available in the Appendix A.

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
