# Peer review of "Measurement of Oxygen Transfer Rate and Specific Oxygen Uptake Rate of h-iPSC Aggregates in Vertical Wheel Bioreactors to Predict Maximum Cell Density Before Oxygen Limitation"

_bioengineering, 2025, doi:10.3390/bioengineering12040332_

Round 1
Reviewer 1 Report
Comments and Suggestions for Authors
The authors have reported Measurement of oxygen transfer rate and specific oxygen up- 2
take rate of h-iPSC aggregates in vertical wheel bioreactors to 3predict maximum cell density before oxygen limitation. The article is well written, yet there are few concerns.
Comments.
The authors need to tighten the abstract
The introduction last para is usually the objective, the last two paragraphs are not very clear. Kindly redo it. This is a important part of the introduction, so you need to make sure that it will well written. The usage of numerous 'we' should be limited. Use passive voice. throughout
The figures lack statistical analysis -Fig 3 and whereever needed need to change.
Add foot note to the Tables.
the figures merely present the statistical data as SD, some kind of PCA analysis should be added.
The presentation of the data is majorly in Tables, kindly bring it to a better analytical presentation that would enrich the paper.
Discussion can be worked on a bit more
Reviewer 2 Report
Comments and Suggestions for Authors
The authors studied the optimal cell culture density for iPSC culture in various 3D bioreactors. The authors measured the oxygen mass transfer coefficient (kLa) for surface aeration under different agitation rates, headspace gas flow rates, and working volume. Thereafter, the authors also measured the specific oxygen uptake rate (sOUR) of iPSC aggregates. From these two key parameters, and guidelines from previous studies, the authors predicted the maximum cell density for iPSCs for different sized bioreactors at recommended agitation rates. This manuscript was well-written, however, it could still be improved by strengthening its discussion.
Comments:
- It would be helpful if the author could provide more background/discussion on the assumptions made behind the sOUR value. Is the sOUR expected to stay constant throughout the multi-day culture? The authors measured the sOUR after 7 days of culture, is it expected to be different when cells are just freshly passaged or when the cells have already experienced multiplying growth?
- How does the calculated maximum cell density accommodate for cell growth over time?
- The authors should specify how data are represented for all measured data, e.g. mean or median, error, number of replicates, etc.
- A reference is missing for line 271 – 274 on the optimal agitation rate
- At Line 136, a reference may be missing for the cell culture method, “… the cells were harvested as previously described …”
- For the supplementary tables, the authors need to label which table refers to which reactor volume, e.g. “Summary of the kLa measured in PBS 3 at a) 2L, b) 3L, …”
- At Line 211, there is a typo, “All O2 inroduced into the …”
- At line 214, there is a typo, “Wit these assumptions …”
Round 2
Reviewer 1 Report
Comments and Suggestions for Authors
Accept